# A Comprehensive Study of the Synthesis, Spectral Characteristics, Quantum–Chemical Molecular Electron Density Theory, and In Silico Future Perspective of Novel CBr_3_-Functionalyzed Nitro-2-Isoxazolines Obtained via (3 + 2) Cycloaddition of (*E*)-3,3,3-Tribromo-1-Nitroprop-1-ene

**DOI:** 10.3390/molecules30102149

**Published:** 2025-05-13

**Authors:** Karolina Zawadzińska-Wrochniak, Karolina Kula, Mar Ríos-Gutiérrez, Bartłomiej Gostyński, Tomasz Krawczyk, Radomir Jasiński

**Affiliations:** 1Department of Organic Chemistry and Technology, Cracow University of Technology, Warszawska 24, 31-155 Cracow, Poland; karolina.zawadzinska@biprotech.com; 2Department of Organic Chemistry, University of Valencia, Dr. Moliner 50, Burjassot, 46-100 Valencia, Spain; m.mar.rios@uv.es; 3Department of Structural Chemistry, Centre of Molecular and Macromolecular Studies Polish Academy of Sciences, Sienkiewicza 112, 90-363 Łódź, Poland; bartlomiej.gostynski@cbmm.lodz.pl; 4Department of Chemical Organic Technology and Petrochemistry, Silesian University of Technology, Krzywoustego 4, 44-100 Gliwice, Poland; tomasz.krawczyk@polsl.pl

**Keywords:** (3 + 2) cycloaddition, nitrile oxide, nitroalkene, molecular mechanism, molecular electron density theory, molecular docking

## Abstract

The search for new heterocyclic compounds with biological potential is one of the current challenges in modern chemistry. Therefore, the comprehensive study of (3 + 2) cycloaddition (32CA) reactions between a series of aryl-substituted nitrile N-oxides (NOs) and (*E*)-3,3,3-tribromo-1-nitroprop-1-ene (TBNP) is carried out. According to the experimental research, in all tested 32CAs, the proper (4*RS*,5*RS*)-3-aryl-4-nitro-5-tribromomethyl-2-isoxazolines are obtained as only one reaction product. In turn, the quantum–chemical MEDT study shows that the creation of heterocycles occur via the polar attack of zwitterionic moderate-nucleophilic NOs to strong electrophilic TBNP. The reactions are realized according to a two-stage, one-step asynchronous mechanism, in which the formation of the O-C(CBr_3_) bond takes place once the C-C(NO_2_) bond is already formed. What is more, the computational analysis confirmed the experimental results. At the end, the obtained 2-isoxazolines were docked to three proteins: gelatinase B, cyclooxygenase COX-1, and Caspase-7. We hope that the presented study will be helpful for searching for the future direction of application for this class of organic compounds.

## 1. Introduction

The application of nitrofunctionalized organic compounds for the synthesis of useful heterocycles is becoming increasingly important [1]. In this context, conjugated nitroalkenes (CNAs) [2,3,4] as well as conjugated nitrodienes (CNDs) [5,6] play a role. Due to the extensive biological properties of CNAs and CNDs, including antibacterial [7,8] and also antifungal [9], these compounds constitute an attractive building block in organic synthesis. In particular, CNAs and CNDs are widely applied via cycloaddition (CA) reactions [10,11]. As a result, the synthesis of nitrofunctionalized carbo- and heterycyclic systems is possible, such as analogs of cyclopentane [12], cyclohexene [13,14], as well as pyrazolines [15,16], pyrrolidine [17,18], and many others.

In continuation of our research on the synthesis of isoxazolines [19,20,21] via the CA protocol [22,23] and their biological activity [24,25], particularly with the application of trihalomethylated nitroalkenes [26], we performed a series of reactions between analogs of aromatic nitrile N-oxides (ANOs) (**1a-e**) as three-atom components (TACs) [27,28] and (E)-3,3,3-tribromo-1-nitroprop-1-ene (TBNP) (**2**). Assuming the single-step reaction mechanism, the cycloaddition of the mentioned addends can theoretically lead to the formation of regioisomeric *4,5-trans*-3-aryl-4-nitro-5-tribromomethyl-2-isoxazoline (path A) (**3a-e**) and *4,5-trans*-3-aryl-4-tribromomethyl-5-nitro-2-isoxazoline (**4a-e**) (path B) (Figure 1).

This choice of reagents was dictated by several reasons. First, ANOs are one of the most popular types of nitrile N-oxides, widely applied in the organic synthesis of heterocyclic systems [29,30]. What is more, due to their interesting electronic properties as well as exceptional zwitterionic type of reactivity, this class of compounds is also commonly investigated in quantum–chemical studies of reaction paths with alkenes such as N-vinyl pyrrol [31], carvone [32], tomentosin [33], as well as benzylideneaniline [34].

On the other hand, trihalomethylated nitroalkenes constitute an important group of compounds due to the presence of two valuable moieties in their structures. In particular, the trihalomethyl –CX_3_ group withdraws electrons at a high rate, thus its presence in compounds can hugely improve molecular properties, such as bioactivity, lipophilicity, and metabolic stability, as well as the ability to penetrate the blood–brain barrier [35,36]. It also should be highlighted that the stability of heterocycles containing a trihalomethyl moiety is even higher compared to compounds without this group [37]. It is also worth mentioning that the presence of the nitro –NO_2_ group creates many possibilities for the transformation of compounds. Thanks to this, it is possible to synthesize many nitrogen-containing connections, like amines, hydroxylamines, oximes, nitriles, diazocompounds, and others, as well as to obtain carbonyl compounds [38,39]. In our research, we decided to use TBNP (**2**), which constitutes an extremely interesting compound both from practical and theoretical points of view. This is mainly related to the fact that the physicochemistry of TBNP (**2**) is still very poorly explored [40,41]. Its properties were described only incidentally in a few studies. In particular, TBNP (**2**) has been tested so far in only a few examples of CA reactions with the participation of nitrones [42,43] as an allylic type of TAC. On the other hand, no papers regarding the activity of TBNP (**2**) in CA processes with the participation of allenyl-type TACs have ever been published.

The presented research is divided into four sections. The first one consists of an analysis of electronic properties for ANOs (**1a-e**) as well as TBNP (**2**) based on the Molecular Electron Density Theory (MEDT) [44,45]. Then, the titled reactions were tested experimentally. Next, the obtained products were isolated and purified. The constitution of these molecules was confirmed based on physicochemical and spectral characteristics. In the third part of this manuscript, the phenomenon of the creation of the final product was analyzed via the Potential Energy Surface (PES) as well as Bonding Evolution Theory (BET) [46,47]. Finally, the obtained 2-isoxazolines were docked to three proteins: gelatinase B, cyclooxygenase COX-1, and Caspase-7 [48,49].

## 2. Results and Discussion

### 2.1. MEDT Study of Electronic Properties of ANOs (1a-e) and TBNP (2)

First of all, we decided to conduct research on the electronic properties of ANOs (**1a-e**) and TBNP (**2**). Thanks to this, it is possible to determine the type of reaction, electron density flux, and also reaction polarity, as well as the most-preferred reaction path from a regioisomeric point of view [50,51].

#### 2.1.1. Study of the Electronic Properties of Used Reagents Based on the ELF and NPA

In order to gain an insight into the reactivity of the reagents involved in the tested reactions of CA, their electronic structure was characterized through the topological analysis of the Electron Localization Function (ELF) [52] together with a Natural Population Analysis (NPA) [53,54] of their charge distribution. Our considerations focused mainly on the electronic structure of the TBNP (**2**) due to the fact that it has not been studied so far using ELF and NPA techniques. In order to show the general electron population in the most significant moiety of ANOs (**1a-e**), the structures of the simplest benzonitrile N-oxide (BNO) (**1**) were analyzed. Thus, ELF basin attractor positions together with some relevant valence basin populations, as well as the proposed Lewis-like structures together with the natural atomic charges are shown in Figure 1, while the ELF basin attractor positions together with the relevant valence basin populations for the remaining ANOs (**1a-e**) are shown in Appendix A.

The ELF topological analysis of the most significant fragment for BNO (**1**) shows the presence of two monosynaptic basins, which are V(O1) and V′(O1), integrating a total electron population of 5.30 e and located at the more nucleophilic oxygen O1 atom; one disynaptic basin, namely V(O1,N2), integrating 1.56 e and associated with an underpopulated O1–N2 single bond; and also two pairs of disynaptic basins, V(N2,C3) and V′(N2,C3), integrating a total electron population of 5.70 e and associated with a somewhat depopulated N2–C3 triple bond (Figure 1). Due to the presence of the N2–C3 triple bond as well as the absence of a lone pair of electrons on the central N2 nitrogen atom, the ELF topological analysis of BNO (**1**) indicated that this TAC can be classified as a propargyl (line) [55] type of zwitterionic electronic structure [56] participating in *zw*-type 32CA reactions [27,57]. The introduction of a substituent at the *para* position of the phenyl ring in BNO (**1**) changes the electron population distribution in all ANOs (**1a-e**), but not in a significant way. In particular, in ANOs (**1a-e**), a total electron population for the oxygen O1 atom is in the range from 5.67 to 5.79 e (Appendix A), corresponding to a higher population compared to the oxygen O1 atom of BNO (**1**). What is more, the total electron population for the bond between N2 and C3 atoms in ANOs (**1a-e**) ranges from 5.91 to 6.02 e (Appendix A), which is connected with the more-populated triplet bond than in BNO (**1**). Regardless of the presented differences, all TACs (**1a-e**) also have zwitterionic electronic structures.

In turn, the ELF topological analysis of the double bond of the alkene moiety in TBNP (**2**) shows the presence of two pairs of disynaptic basins, V(C4,C5) and V′(C4,C5), integrating a total electron population of 3.57 e (Figure 1). The presence of these disynaptic basins is associated with the depopulated C4-C5 double bond, which is an effect of the presence of two electron-withdrawing -CBr_3_ and -NO_2_ groups situated in the vicinal position of the ethylene moiety in TBNP (**2**). Based on the ELF analysis, we proposed Lewis-like structures.

NPA studies were also carried out. The NPA analysis of BNO (**1**) indicated that the oxygen O1 atom was strongly negatively charged by −0.40 e, while both nitrogen N2 and carbon C3 atoms were moderately positive charged by +0.21 e and +0.20 e, respectively (Figure 1). The presented charge distribution is a consequence of the polarization of the C–N–O framework toward the more electronegative oxygen O1 atom.

On the other hand, the NPA analysis of TBNP (**2**) indicates a significant negative charge of the carbon C4 atom by −0.24 e, while the carbon C5 atom is only weakly negatively charged by −0.08 e (Figure 1). The observed difference in charge distributions is a consequence of the presence of the electron-withdrawing moiety with different effects, namely nitro as well as tribromomethyl groups, in the structure of alkene (**2**).

#### 2.1.2. CDFT View of Global and Local Reactivity

In the subsequent part of this research concerning the reactivity of the used reagents, the Conceptual Density Functional Theory (CDFT) were performed. Overall, the CDFT is an important approach to understand the reactivity of molecules. The theory connects well-established chemical concepts, like electronic chemical potential, μ, and chemical hardness, η, with the electronic structure of a molecule [58]. In turn, the previously mentioned descriptors are useful to determine the global electronic properties of substrates, such as global electrophilicity, ω, and global nucleophilicity, N [59,60]. As a result, it is possible to classify reagents as electrophiles or nucleophiles in studied reactions, as well as indicate the local electronic properties of a molecule [61,62]. The global reactivity indices for ANOs (**1a-e**) and TBNP (**2**) are presented in Table 1.

The application of the CDFT is strongly dependent on the processes polarity [63] and can be applied only for polar reactions. The preliminary determination of the polarity of tested reactions is necessary for conducting further tests. For this purpose, the electronic chemical potential, μ, was analyzed. The electronic chemical potential, μ, of TBNP (**2**), μ = −5.71 eV, is significantly lower compared to the tested ANOs (**1a-e**), where the μ value ranges from −3.99 to −4.78 eV. This means that the electron density flux will take place from ANOs (**1a-e**) to TBNP (**2**) (Figure 2). So, the considered reaction between the analyzed reagents should be classified as the forward electron density flux (FEDF) [50]. According to the older Sustman’s classification, the title processes should be defined as normal-electron-demand (NED) cycloadditions [50]. What is more, the difference of global electrophilicities between ANOs (**1a-e**) and TBNP (**2**) suggests that the analyzed cycloaddition should be treated as a weak polar process (Table 1 and Figure 2).

Reactivity descriptors were used to determine global electronic properties. Thus, the computed global electrophilicity [64,65] ω indices of the tested ANOs (**1a-e**) are 0.91 eV for **1a** (-OCH_3_), 0.90 eV for **1b** (-CH_3_), 0.94 eV for **1c** (-F), 1.07 eV for **1d** (-Cl), and 1.40 eV for **1e** (-NO_2_), while the calculated global nucleophilicity [64,66] N indices for these compounds are 2.39 eV for **1a** (-OCH_3_), 2.43 eV for **1b** (-CH_3_), 2.28 eV for **1c** (-F), 2.07 eV for **1d** (-Cl), and 1.94 eV for **1e** (-NO_2_) (Table 1). In general, the presence of electron donor groups (EDGs) determines the decrease in global electrophilicity. Next, the presence of electron-withdrawing groups (EWGs) determines the increase in global electrophilicity. As expected, the nature of the substituent has exactly the opposite effect on the values of the nucleophilicity indices. The obtained values of reactivity descriptors lead also to the conclusion that, due to a similar ω index, within the Domingo scale [58], ANO (**1a-d**) should be classified as a moderate electrophile, while ANO (**1e**) has a slightly higher ω index; therefore, within the same scale [58], it can be classified as a strong electrophile. On the other hand, all considered ANOs (**1a-e**) may be classified as moderate nucleophiles.

In turn, for TBNP (**2**), the computed global electrophilicity [64,65] ω index is 1.83 eV, and this value is considerably higher compared to the ω indices for all ANOs (**1a-e**) (Table 1). The calculated global nucleophilicity [64,66] N index for this nitroalkene (**2**) is 0.67 eV, and this value is lower compared to the ω indices for all ANOs (**1a-e**) (Table 1). These values lead to the conclusion that TBNP (**2**) can be classified as a strong electrophile and marginal nucleophile in a polar reaction.

Overall, in all the considered reactions, it can be assumed that all ANOs (**1a-e**) will participate as nucleophilic agents, while TBNP (**2**) will play the role of an electrophilic component. This conclusion is valuable information for future considerations about local reactivity. In reactions with non-symmetric reagents, regioselectivity can be defined as the interaction between the most electrophilic center of the electrophile and the most nucleophilic center of the nucleophile [61,62]. Therefore, to characterize the most electrophilic centers for TBNP (**2**) and the most nucleophilic centers for ANOs (**1a-e**), the electrophilic P_k_^+^ and nucleophilic P_k_^−^ Parr functions were investigated (Figure 3).

The analysis of nucleophilic P_k_**ˉ** Parr functions of ANOs (**1a-e**) shows that, for all nitrile N-oxides, regardless of the substituent at the *para* position of the phenyl ring, the most nucleophilic center of this specie is located on the oxygen O1 atom of the TAC moiety. The values for P_O1_^−^ are 0.47 for **1a** (-OCH_3_), 0.47 for **1b** (-CH_3_), 0.49 for **1c** (-F), 0.52 for **1d** (-Cl), and 0.54 for **1e** (-NO_2_) (Figure 3). On the other hand, the analysis of the electrophilic P_k_^+^ Parr function of TBNP (**2**) indicates that the C5 carbon atom is the most electrophilic center of this species, presenting the value P_C5_^+^ = 0.28 (Figure 3). The obtained results based on the CDFT lead to the conclusion that all reactions of ANOs (**1a-e**) with TBNP (**2**) are determined by the nucleophilic attack of the oxygen O1 atom of TACs (**1a-e**) on the electrophilic carbon C5 atom of nitroalkene (**2**), directly connected with the tribromomethyl group. Consequently, the creation of (4*RS*,5*RS*)-3-aryl-4-nitro-5-tribromomethyl-2-isoxazolines (**3a-e**), according to channel A (Figure 1), as the most favorable regioisomer, is more probable.

### 2.2. Experimental Study of (3 + 2) Cycloaddition Reactions Between ANOs (1a-e) and TBNP (2)

#### 2.2.1. Protocol Synthesis Details of Cycloaddition Components (1a-e) and (2)

At the beginning of the experiment, a synthesis of reagents was carried out. The precursors for ANOs (**1a-e**) were prepared according to a two-step procedure. First, the corresponding aromatic aldehydes were converted into the respective oximes via the procedure, including reactions with hydroxylamine hydrochloride in ethanol. In the next step, in order to obtain the hydroxymoyl chlorides as precursors of the ANOs (**1a-e**), a chlorination process of the oximes with N-chlorosuccinimide (NCS) in N,N-dimethylformamide (DMF) was performed. Finally, ANOs (**1a-e**) were generated in situ directly from their respective hydroxymoyl chlorides in the main reaction mixtures (Figure 2) [67].

TBNP (**2**) was obtained via a three-step protocol. First of all, the condensation reaction between nitromethane and tribromoacetaldehyde in the presence of potassium carbonate was carried out. Next, the obtained alcohol was converted into the respective ester via a reaction with acetyl chloride. The final TBNP (**2**) was synthesized through nitroester decomposition in the presence of the sodium carbonate in benzene (Figure 3) [40,41].

#### 2.2.2. Regio- and Stereochemistry of the Reaction Between ANOs (1a-e) and TBNP (2)

In the next part of this research, the synthetic study of the reaction between ANOs (**1a-e**) and TBNP (**2**) is performed. It should be highlighted that the considered 32CA reactions can yield two regioisomeric cycloadducts from a theoretical point of view (Figure 1). Based on the information about similar reactions of (3 + 2) cycloaddition with the participation of trihalogenated nitrocompounds and allenyl-type TACs [15,26,43], we decided to conduct the series of reactions between ANOs (**1a-e**) and TBNP (**2**) in the presence of potassium carbonate in THF at room temperature.

The reaction progress of all considered 32CAs was monitored by the TLC technique. In the previously mentioned conditions, it was found that the conversion of addents occurred in about 24 h. Furthermore, TLC analysis indicated that, in each reaction mixture, only one product was formed as a solid.

The synthesized products were isolated from the reaction mixture via a sequence of column chromatography and then purified by crystallization. The constitution of the final compounds was identified based on physicochemical and spectral characteristics.

In order to confirm the structure of the obtained compounds, spectral physicochemical analyses were performed and spectral characteristics, such as HR-MS, IR, ^1^HNMR, and ^13^C NMR, were obtained. The full spectral characteristics are included as Appendix A.

First, HR-MS analysis was performed. All HR-MS spectra present pseudo-molecular ions with *m*/*z*, which are compatible with the molecular formula of the expected products. Thus, it can be concluded that the obtained compounds will be one of the two theoretical regioisomers of isoxazolines (**3a-e**) or (**4a-e**) (Figure 1).

The structures of the hypothetical products were also confirmed by IR analysis. In particular, in all spectra, it was possible to identify characteristic bands for the NO_2_ [5,6], -N-O- [22,23], and -C=N- [68,69] moieties for the heterocyclic ring, and C-Br bond [42] vibrations were detected in the IR spectrum.

In order to determine the regioisomerism of the formed isoxazolines (**3a-e**) or (**4a-e**), the analysis of NMR was carried out. The distribution of signals in the ^1^H NMR spectra allowed us to conclude that in all considered 32CA reactions between ANOs (**1a-e**) and TBNP (**2**), the synthesized compounds are (4*RS*,5*RS*)-3-aryl-4-nitro-5-tribromomethyl-2-isoxazolines (**3a-e**) (Figure 1). In particular, protons associated with the isoxazoline ring form the AX spin system are characteristic of this class of heterocyclic compounds [26]. What is more, it should be noted that, based on the values of J_4,5_ isoxazoline ring-coupling constants in final products, the relative position of both tribromomethyl as well as nitro groups is in line with the theoretical assumption in the *trans* configuration [6,43]. On the other hand, the localization of proton doublets 4 and 5 confirms the reaction regioselectivity. In the case of hypothetical 5-nitro-2-isoxazoline, the H5 proton signal is expected in to be in a relatively weaker field, whereas the H4 proton signal will be localized in the strongest field.

### 2.3. Quantum–Chemical Study of the Reaction Course

Both the analysis of the electronic structures of reagents based on the ELF, NPA, and CDFT, as well as the experimental consideration indicate that, during all 32CA reactions between ANOs (**1a-e**) and TBNP (**2**), the synthesized compounds are (4*RS*,5*RS*)-3-aryl-4-nitro-5-tribromomethyl-2-isoxazolines (**3a-e**) (Figure 1), irrespective of the substituent in the *para* position of the phenyl ring in TACs (**1a-e**). To better understand the molecular mechanism of the tested reactions, analyses based on PES as well as BET approaches were carried out.

#### 2.3.1. Exploration of the PES of Cycloadditions of ANOs (1a-e) with TBNP (2)

In the third part of this manuscript, the phenomenon of the creation of the final product is analyzed via Potential Energy Surface (PES) as well as the Bonding Evolution Theory (BET). Our theoretical study of the tested 32CA reactions of ANOs (**1a-e**) with TBNP (**2**) started from the analysis of PES. For this purpose, thermodynamic and kinetic parameters were determined and are presented in Table 2 and Figure 4. The solvent effect was included via the polarizable continuum model (PCM) [70,71,72] for the THF solution.

The computed data show that the first stage of all studied 32CAs reactions is related to the formation of two independent transition states (namely **TS_A_** and **TS_B_**) (Figure 4). The IRC results for all the obtained TSs indicate the presence of only one imaginary eigenvalue in Hessian. What is more, the IRC-computed TSs are directly connected with the energy minimum of substrates and products in any reaction pathway. These results confirm the absence of more energy minimum values connected with the presence of other critical structures. Therefore, it should be concluded that the mechanism of the tested 32CAs can be considered as *one-step* reactions.

The obtainment of TSs is associated with the increase in the Gibbs free energy of the process in the range of ca. 20–25 kcal‧mol^−1^ (Table 2). According to Eyring parameters, it can be noted that in all the considered 32Cas, the formation of (4*RS*,5*RS*)-3-aryl-4-nitro-5-tribromomethyl-2-isoxazolines (**3a-e**) is the most preferable product from a kinetic point of view (Figure 4). The activation barriers for path A are ca. 2–3 kcal‧mol^−1^ higher compared to the competitive path B (Table 2). However, due to insignificant differences in the activation barriers, both of the theoretical reaction paths can be considered possible from a kinetic point of view. The obtained activation energies are characteristic of polar *zw*-type 32CA reactions [56]. Interestingly, a correlation between the activation energies and the nucleophilicities can be observed—the less the nucleophilic ANO, the higher the activation energy.

The step related to the formation of the final product is associated with a significant decrease in Eyring parameters (Figure 4). In particular, the enthalpies of reaction are above −40 kcal‧mol^−1^ (Table 2). Based on this information, it may be noted that all considered 32CAs are exothermic.

#### 2.3.2. Analysis of Critical Structures for the Studied Reaction of ANOs (1a-e) with TBNP (2)

In order to determine the synchronicity of the considered 32CA reactions of ANOs (**1a-e**) with TBNP (**2**), a comprehensive analysis of all critical structures located in the reaction paths was carried out. The results include key parameters, such as the interatomic distance between reaction centers for significant structures (r), the development of a new single bond (l), and an asymmetry index, Δl, which are presented in Table 3 [73,74]. To confirm the moderate polarity of the tested reactions, the values of Global Electron Density Transfer (GEDT) [45,75] for all TSs were also determined. The visualization of key critical structures for reaction between ANO (**1a**) and TBNP (**2**) is shown Figure 5.

As noted in the previous section, the first stage of all considered 32CAs between ANOs (**1a-e**) and TBNP (**2**) is the formation of TSs (Table 3). Interatomic distances for O1–C5 and also C3–C4 bonds in these TSs are in a typical range for similar reactions of cycloadditions [15,42,55]. In particular, for all TSs located on path A, the lengths of the formed bonds of O1–C5 are smaller, while the bonds of C3–C4 are bigger, in comparison to the same bonds in TSs located on path B (Table 3). This observation is in agreement with the most favorable interactions between the most nucleophilic and electrophilic centers [51,76]. What is more, according to Δl parameters quantifying the bond formation extent in reference to the products’ distances, it should be concluded that the all A paths for the obtained (4*RS*,5*RS*)-3-aryl-4-nitro-5-tribromomethyl-2-isoxazolines (**3a-e**) are definitely more synchronous (Δl practice 0.00) in comparison to the competitive regioisomeric B paths (Δl about 0.10) (Table 3). Thus, the tested 32Cas, according to path B, can be considered as moderately asynchronous reactions, while according to path A, they can be considered as slightly asynchronous reactions [73,74].

At the end, the GEDT parameters computed at the ANO frameworks for each TS are very close, ca. 0.10 e (Table 3), which confirms the somewhat polar character of all tested 32CAs and the FEDF classification of the reactions [50].

#### 2.3.3. BET Analysis for the Studied Reaction of ANOs (1a-e) with TBNP (2)

At the end, a thorough analysis of the bond formation processes according to the BET was carried out. In general, the BET combines the ELF [52] with the catastrophe theory (CT) [46,47] and allows us to predict the electron density rearrangement along a reaction path. As a result, following the bonding changes associated with a given molecular mechanism is possible [77]. In this context, the BET analysis for the 32CA reaction between ANO (**1a**) with TBNP (**2**) for path A, leading to an experimentally synthesized product, (4*RS*,5*RS*)-3-aryl-4-nitro-5-tribromomethyl-2-isoxazoline (**3a**), was carried out. In Table 4, electron populations of the most relevant ELF valence basins as well as O1-C5 and C3-C4 distances for **TS_3a_**, **3a**, and also **S1-S4** structures involved in the formation of isoxazoline ring are presented, while in Figure 6, the visualization of this process, including the most significant attractor positions of the ELF basins for **S1-S4** structures, is shown.

The BET analysis of the bonding changes along the 32CA reaction between ANO (**1a**) and TBNP (**2**) according to path A shows that, in **TS_3a_**, both the N2-C3 and C4-C5 bonding regions have been depopulated with respect to the separated reagents. In particular, for the N2-C3 bond, the electron population in the ANO (**1b**) is 6.01 e, associated with a triple bond, while **TS_3a_** is significantly reduced to 4.40 e (Table 4), characteristic of an overpopulated double bond. On the other hand, for the C4-C5 bond, the electron population in the TBNP (**2**) is 3.57 e (Figure 1), associated with an underpopulated double bond (Figure 1), while **TS_3a_** is slightly reduced to 3.44 e (Table 4), but is still characteristic of an underpopulated double bond.

The approaching of structures of ANO (**1a**) and TBNP (**2**) lead to structure S1. For this structure, the shortening of O1-C5 and C3-C4 atomic distances is noticeable, especially for C3-C4 (Table 4). Simultaneously, the electron populations of N2-C3 and C4-C5 bonds are reduced, in particular, for the C4-C5 bond, the reduction is ca. 20% (Table 4). At the same time, the creation of two new monosynaptic basins at the C3 and C4 carbon atoms is noticeable (Figure 6), integrating, respectively, 0.50 e for V(C3) and 0.24 e for V(C4) (Table 4). This observation can be identified as the beginning of the first C3-C4 single-bond formation in the isoxazoline ring by the creation of two C3 and C4 *pseudoradical* centers [78].

Further conversion of substrate (**1a**) TBNP (**2**) leads to structure S2. For this structure, the shortening of O1-C5 and C3-C4 atomic distances is observed (Table 4). The monosynaptic basins, located on the C3 and C4 carbon atoms, are merged (Figure 6), presenting in total the electron population of 1.29 e for the V(C3,C4) disynaptic basin (Table 4), characteristic of an underpopulated single bond. As a result, the first, new, single-bond C3-C4 is created. At the same time, a new monosynaptic basin at the C5 carbon atom is noticeable (Figure 6), with an electron population of 0.23 e for V(C5) (Table 4), which is associated with a C5 *pseudoradical* center [78]. This fact suggests the beginning of the formation of the second single bond in the isoxazoline ring. Both of these observations can be related to the further explicitly noticeable depopulation of the C4-C5 bond.

In the subsequent S3 structure, the electron population of the C5 *pseudoradical* center slightly increases to 0.32 e for V(C5), while a new monosynaptic basin, located on the O1 oxygen atom, appears (Figure 6), integrating 0.37 e with V″(O1) (Table 4). Simultaneously, the electron population of other monosynaptic bases, namely V(O1) and V′(O1), is evidently reduced.

In the last structure, S4, the most significant observation is related to the merging of two V″(O1) and V(C5) monosynaptic basins (Figure 6), presenting in total an electron population of 0.86 e for the V(O1,C5) disynaptic basin (Table 4), characteristic of a strongly underpopulated single bond. In a way, the second, new, single-bond O1-C5 is created. Then, the whole structure of product **3a** presents noticeable changes. In particular, both O1-C5 and C3-C4 are reduced to values characteristic of sterically crowded isoxazoline rings (Table 4) [30,79].

Interestingly, the analysis of the bonding changes for the mentioned atoms along the 32CA reaction shows that the evolution of the bond formation is considerably asynchronous. Note that the second O1-C5 bond is formed at a d(O1-C5) of 1.72 Å when the first C3-C4 bond is almost formed by up to 84%, characteristic of a *two-stage*, *one-step* mechanism [80]. This observation is consistent with the usual distances for C-C and C-O bond formations [75] and the geometrical asymmetry of the TSs (Table 3). What is more, the electron population of O1-C5 and C3-C4 bonds has noticeably increased. For the C3-C4 bond, the electron population is higher ca. 25%, presenting in total the electron population of 2.02 e for V(C3,C4) (Table 4), characteristic of a single bond. On the other hand, for the O1-C5 bond, the electron population is lower ca. 50%, presenting in total the electron population of 1.64 e for V(O1,C5) (Table 4), characteristic of a significantly polarized single bond. The increase in both new single bonds is due to the depopulation of other regions, in particular, the monosynaptic basins located on the O1 oxygen atom, namely V(O1) and V′(O1), by over 5% (Table 4). It is interesting to note that the C3-C4 single bond is formed prior to the O1-C5 one involving the most nucleophilic and electrophilic centers. Thus, in this case, the first single-bond formation is not between the most electronically activated atoms, suggesting that other factors may play a relevant role in determining the initial interaction. This finding was also observed in the 32CA reactions of the same family of ANOs with chlorated TCNP derivatives [26].

### 2.4. Molecular Docking Consideration of the Obtained (4RS,5RS)-3-aryl-4-Nitro-5-Tribromomethyl-2-Isoxazolines (3a-e)

Both synthetic studies and theoretical considerations based on the MEDT clearly confirm that, during all tested 32CAs between ANOs (**1a-e**) and TBNP (**2**), (4*RS*,5*RS*)-3-aryl-4-nitro-5-tribromomethyl-2-isoxazolines (**3a-e**) as the only reaction products were obtained (Figure 1). It is well known that isoxazolines and their analogues are very useful compounds, mainly in medicine as various pharmacophores [81,82,83,84,85,86,87]. Therefore, at the end, we decided to perform a molecular docking study for the obtained heterocycles (**3a-e**).

For this purpose, the optimized geometries of the obtained (4*RS*,5*RS*)-3-aryl-4-nitro-5-tribromomethyl-2-isoxazolines (**3a-e**) were docked to the same proteins that were used in the previous work on their very close structural relatives [24], namely gelatinase B (RSCB PDB ID: 4XCT), which is a metalloproteinase possessing a high capacity for the regulation of cytokine and chemokine activity [88]; cyclooxygenase COX-1 (RSCB PDB ID: 3KK6), which is an enzyme, partaking in the catalytic synthesis of prostaglandins that cause pain and inflammation [89]; as well as Caspase-7 (RSCB PDB ID: 4ZVT), which is a protein acting as an apoptosis executioner [90].

The docking results interpretation was based on three different parameters delivered by Gnina: *CNN pose score, CNN affinity* (Table 5), as well as *affinity* values (Table 6). The *CNN pose score* is the dimensionless neural network measure of likelihood that the ligand is in the position close to what its experimental position would be, although a pose score greater than 0.8 indicated at least 56% probability that the ligand has a RMSD < 2 Å compared to the experimental value. In turn, the *CNN affinity* (presented in pK units) is a dimensionless neural network measure of the binding ability for a given pose, hence a stronger binder will exhibit larger values for this parameter. At the end, the a*ffinity* is the “classic” scoring energy outcome (presented in kcal‧mol^−1^), so as a better binder, it is supposed to exhibit lower values for this parameter. As a main model of considered structures, the appropriate (4*R*,5*R*)-isomers of 2-isoxazolines (**3a-e**) were applied.

The results presented in Table 5 suggest that the optimal binding mode for (4*R*,5*R*)-isomers of 2-isoxazolines (**3a-e**) is the 3.5 Å flexible distance. Therefore, these results were used for the presentation of the binding affinity, and to complete this research, the docking of (4*S*,5*S*)-isomers 2-isoxazolines (**3a-e**) was calculated.

From the presented results in Table 5 and Table 6, it can be concluded that the investigated 2-isoxazolines (**3a-e**) adopt conformations that are characteristic of a moderate-to-good binding affinity, among which the best binders to 4XCT, 3KK6, and 4ZVT are the **3d** and **3a** for (4*R*,5*R*)-isomers as well as **3b** and **3d** for (4*S*,5*S*)-isomers. Additionally, molecule **3a** has notably stronger binding activity to 3KK6 than 4ZVT and 4XCT, presenting the best docking score of −9.25 kcal‧mol^−1^ for the (4*R*,5*R*)-isomer, with other compounds presenting better affinities scores; yet, though both present notably lower values for both remaining parameters in general and for the (4*S*,5*S*)-isomer, the best target seems to be 3KK6.

Based on the obtained results of affinities (Table 6), the visualizations, including structures of exemplary 2-isoxazolines (4*R*,5*R*)-**3d** docked within 4XCT as well as (4*R*,5*R*)-**3a** within 3KK6 and also within 4ZVT, together with their native ligands were prepared and are presented in Figure 7, Figure 8 and Figure 9.

As can clearly be seen, Figure 7 depicts a metalloprotein with a (4*R*,5*R*)-**3d** ligand docked in the vicinity of the metal (Zn) cation. Distances of the O-Zn (oxygen–zinc) ligand are elongated in comparison to the native ligand (3.5 and 4.3 Å vs. 2.2 and 2.3 Å), yet visibly strong enough to lower the binding energy and make this ligand the most favorable energetically.

In turn, Figure 8 presents the (4*R*,5*R*)-**3a** ligand buried deeply in the active pocket, in the direct vicinity of the native ligand. Yet, this structure is devoid of strong interactions, scoring the second-best score. Finally, in Figure 9, a relatively loosely bonded (4*R*,5*R*)-**3a** ligand can be seen. It is not inserted as deep as the native ligand, and its bonding affinity is the lowest of the three.

In conclusion, gelatinase B seems to be the best target for 2-isoxazoline (4*R*,5*R*)-**3d**, which has the most promising docking scores, whereas 2-isoxazoline (4*R*,5*R*)-**3a** has the best score for both cyclooxygenase COX-1 and Caspase-7. In these cases, the docking scores are significantly lower, but the obtained results also suggest that 2-isoxazoline (4*R*,5*R*)-**3a** is more valid ligand for cyclooxygenase COX-1 than for Caspase-7 (Table 5 and Table 6).

On the other hand, in the case of (4*S*,5*S*)-isomers (**3a-e**), the energy scores are different. In particular, the best binder to gelatinase B turned out to be 2-isoxazoline (4*S*,5*S*)-**3d**, while for COX-1 and Caspase-7, the best scorers were 2-isoxazolines (4*S*,5*S*)-**3b** and again (4*S*,5*S*)-**3d** (Table 5 and Table 6).

## 3. Materials and Methods

### 3.1. Materials

Commercially available (Sigma–Aldrich, Szelągowska 30, 61-626 Poznań, Poland) reagents and solvents were used. All solvents were tested with high-pressure liquid chromatography before use.

The components of the cycloaddition were prepared according to the procedures described in the literature. In particular, TBNP (**2**) was obtained via a three-step method, starting from nitromethane and tribromoacetaldehyde (bromal) [40,41]. In turn, the ANOs (**1a-e**) were generated in situ from hydroxamoyl chlorides, obtained via the chlorination of oximes according to the known procedures [67].

### 3.2. Cycloaddition Between ANOs (1a-e) and TBNP (2)—General Procedure

An Erlenmeyer flask with a magnetic stirrer, containing 10 cm^3^ of THF, was placed in an ice bath. After cooling the mixture to a temperature of 0 °C, 0.0030 mole of (*E*)-3,3,3-tribromo-1-nitroprop-1-ene and 0.0025 mole of hydroxymoyl chloride were added and stirred for 10 min. Then, 0.0015 mole of K_2_CO_3_ was dosed in small portions over a 30 min period. Afterward, the ice bath was removed. Change in mixture color and turbidity were observed. The mixture was left for 24 h with constant stirring, and this time was monitored via TLC (SiO_2_; hexane:ethyl acetate mix. Hex:EtOAc 9:1 *v*/*v*). The solvent was evaporated, and the remaining solid was mixed with diethyl ether and filtrated to remove insoluble side products. The ether was removed under vacuum, and the remaining crude product was washed with light petroleum ether. The isolation and future purification of the reaction products were performed via column chromatography or/and crystallization. As a result, appropriate racemic mixtures of (4*RS*,5*RS*)-3-aryl-4-nitro-5-tribromomethyl-2-isoxazoline were obtained and analyzed.

(4*RS*,5*RS*)-3-(4-methoxyphenyl)-4-nitro-5-tribromomethyl-2-isoxazoline (**3a**): isolation and purification process: column chromatography (hexane:ethyl acetate mix. Hex:EtOAc 9:1 *v*/*v*) and future crystallization (diethyl ether:cyclopentane mix. Et_2_O:CyP 5:5 *v*/*v* on cold); yield 55%; m.p. 112.4 °C (white crystal solid); R_f_ = 0.59; UV-Vis (MeOH): λmax [nm] 268; FT-IR (ATR): υ [cm^−1^] 1606 (>C=N− 2-isox. ring), 1580 asym. and 1365 sym. (−NO_2_), 1258 (~C-O-N= 2-isox. ring), 821 (~C-Br); ^1^H NMR (400 MHz, CDCl_3_): δ [ppm] 7.66–7.62 (m, 4H); 6.28 (d, 1H, J = 3.91 Hz); 5.86 (d, 1H, J = 3.92 Hz); 3.86 (s, 3H); ^13^C NMR (100 MHz, CDCl_3_): δ [ppm] 162.4; 162.3; 151.1; 128.7; 126.9; 112.2; 95.7; 93.9; 55.5; HR-MS (ESI−): calculated for C_11_H_9_N_2_O_4_Br_3_ [M−H]^−^ = 468.8034, found 468.8055.

(4*RS*,5*RS*)-3-(4-methylphenyl)-4-nitro-5-tribromomethyl-2-isoxazoline (**3b**): isolation and purification process: column chromatography (hexane:ethyl acetate mix. Hex:EtOAc 9:1 *v*/*v*) and future crystallization from ethanol; yield 51%; m.p. 86.5 °C (white crystal solid); R_f_ = 0.47; UV-Vis (MeOH): λmax [nm] 265; FT-IR (ATR): υ [cm^−1^] 1605 (>C=N− 2-isox. ring), 1578 asym. and 1366 sym. (−NO_2_), 1259 (~C-O-N= 2-isox. ring), 819 (~C-Br); ^1^H NMR (400 MHz, CDCl_3_): δ [ppm] 7.77–7.72 (m, 4H); 6.30 (d, 1H, J = 4.21 Hz); 5.85 (d, 1H, J = 4.22 Hz); 2.41 (s, 3H); ^13^C NMR (100 MHz, CDCl_3_): δ [ppm] 154.6; 151.5; 142.5; 130.2; 127.0; 122.4; 95.7; 70.2; 21.5; HR-MS (ESI−): calculated for C_11_H_9_N_2_O_3_Br_3_ [M-H]^−^ = 452.8085, found 452.8108.

(4*RS*,5*RS*)-3-(4-fluorophenyl)-4-nitro-5-tribromomethyl-2-isoxazoline (**3c**): isolation and purification process: column chromatography (hexane:ethyl acetate mix. Hex:EtOAc 9:1 *v*/*v*) and future crystallization from methanol; yield 56%; m.p. 111.3 °C (white crystal solid); R_f_ = 0.46; UV-Vis (MeOH): λmax [nm] 258; FT-IR (ATR): υ [cm^−1^] 1601 (>C=N− 2-isox. ring), 1566 asym. and 1361 sym. (−NO_2_), 1249 (~C-O-N= 2-isox. ring), 1231 (~Ar-F), 817 (~C-Br); ^1^H NMR (400 MHz, CDCl_3_): δ [ppm] 7.82–7.78 (m, 4H); 6.29 (d, 1H, J = 3.95 Hz); 5.90 (d, 1H, J = 3.96 Hz); ^13^C NMR (100 MHz, CDCl_3_): δ [ppm] 165.9; 163.4; 150.6; 129.2; 121.6; 116.8; 95.9; 94.0; HR-MS (ESI−): calculated for C_10_H_6_N_2_O_3_Br_3_F [M−H]^−^ = 456.7834, found 456.7850.

(4*RS*,5*RS*)-3-(4-chlorophenyl)-4-nitro-5-tribromomethyl-2-isoxazoline (**3d**): isolation and purification process: column chromatography from chloroform and future crystallization from petroleum ether on cold (greenish–yellowish semi-solid); yield 59%; R_f_ = 0.51; UV-Vis (MeOH): λmax [nm] 262; FT-IR (ATR): υ [cm^−1^] 1591 (>C=N− 2-isox. ring), 1557 asym. and 1354 sym. (−NO_2_), 1252 (~C-O-N= 2-isox. ring), 819 (~C-Br), 556 (~Ar-Cl); ^1^H NMR (400 MHz, CDCl_3_): δ [ppm] 7.76–7.70 (m, 4H); 6.27 (d, 1H, J = 3.87 Hz); 5.91 (d, 1H, J = 3.86 Hz); ^13^C NMR (100 MHz, CDCl_3_): δ [ppm] 168.0; 167.5; 153.0; 129.7; 125.1; 117.4; 95.9; 93.5; HR-MS (ESI−): calculated for C_10_H_6_N_2_O_3_Br_3_Cl [M−H]^−^ = 472.7539, found 472.7547.

(4*RS*,5*RS*)-3-(4-nitrophenyl)-4-nitro-5-tribromomethyl-2-isoxazoline (**3e**): isolation and purification process: column chromatography (chloroform:acetone mix. CHCl_3_:Me_2_CO 8:2 *v*/*v*) (orange oil); yield 50%; R_f_ = 0.42; UV-Vis (MeOH): λmax [nm] 255; FT-IR (ATR): υ [cm^−1^] 1598 (>C=N− 2-isox. ring), 1552 asym. and 1344 sym. (−NO_2_), 1253 (~C-O-N= 2-isox. ring), 850 (~C-Br); ^1^H NMR (400 MHz, CDCl_3_): δ [ppm] 7.92–7.86 (m, 4H); 7.04 (d, 1H, J = 4.25 Hz); 6.57 (d, 1H, J = 4.26 Hz); ^13^C NMR (100 MHz, CDCl_3_): δ [ppm] 156.4; 153.9; 149.5; 133.4; 130.7; 123.7; 101.0; 90.1; HR-MS (ESI−): calculated for C_10_H_6_N_3_O_5_Br_3_ [M−H]^−^ = 483.7779, found 483.7796.

### 3.3. Analytical Techniques

For monitoring the reaction’s progress, thin-layer chromatography (TLC) was performed using aluminum plates with silica (unmodified layers) as the standard procedure in the case of nitro-containing organic compounds [15,16,17], as the eluent hexane:ethyl acetate mix (Hex:EtOAc 9:1 *v*/*v*) was applied. The plates were developed by iodine treatment. The melting points were determined with the *Boetius PHMK 05* apparatus and were not corrected. HR-MS measurements were acquired on a mass spectrometer *Waters Xevo G2 QTof* (ESI ionization) tandem with liquid chromatography (Waters Acquity UPLC BEH Shield 18-RP column: 1.7 µm, 2.1 mm × 100 mm; follow rate: 0.2 mL/min; eluent:acetonitrile:water mix ACN:H_2_O 9:1 *v*/*v*). FT-IR spectra were derived from the *Thermo Scientific Nicolet IS 10* spectrophotometer with *ID7* Attenuated Total Reflectance (ATR). All samples were analyzed in solid form. The spectra were recorded for the range of 500–4000 cm^−1^. ^1^H NMR (400 MHz) and ^13^C NMR (100 MHz) spectra were recorded with an Agilent NMR spectrometer. All spectra data were obtained in the deuterated chloroform CDCl_3_ (visible at 7.27 ppm for ^1^H NMR and at 77.00 ppm for ^13^C NMR) solutions, and the chemical shifts (δ) are expressed in ppm, while the J-couplings (J) are presented in Hz. TMS was used as an internal standard. Data are reported as s = singlet, d = doublet, t = triplet, dd = doublet of doublets, and m = multiplet. UV-Vis spectra were determined in methanolic solutions for the 190–500 nm range with a UV-5100 Biosens spectrometer. The maximum absorption was detected below 1 AU of absorbance and adhered to the Beer–Lambert law.

### 3.4. Computational Details

All computations were performed using the Gaussian 16 package [91] in the Ares computer cluster of the CYFRONET regional computer center in Cracow, as well as in the server of the MEDT research group at the University of Valencia. DFT calculations were performed using the hybrid ωB97X-D functional [92], which includes long-range exchange (denoted by X) correction as well as the semiclassical London-dispersion correction (indicated by suffix-D). The standard 6-311G(d,p) [93] basis set was used, which includes d-type polarization for second-row elements and p-type polarization functions for hydrogen atoms. For calculations, the MEDT approach was used as a common tool in the evaluation of molecular mechanisms of reactions with participation of both organic as well as inorganic compounds [94,95,96], especially for cycloaddition processes [97,98,99]. Calculations of all critical structures were performed at temperature T = 298 K and pressure p = 1 atm in a gas phase. All localized stationary points were characterized using vibrational analysis. It was found that starting molecules as well as products had positive Hessian matrices.

The electronic structures of ANOs (**1a-e**) as well as TBNP (**2**) were characterized by the Electron Localization Function (ELF) [52] and the Natural Population Analysis (NPA) [53,54]. Analyses of global electronic properties of reactants were performed according to Domingo’s recommendations [57,58,64]. Electrophilic Parr functions, Pk^+^, and nucleophilic Parr functions, Pk^−^, were obtained from the Atomic Spin Density (ASD) of the reagents’ radical ions [61,62].

The Berny method was used for TS optimizations [100,101]. The TSs were characterized through frequency analysis, presenting only one imaginary frequency. The intrinsic reaction coordinate (IRC) paths [102] were computed to find the unique connection between the TSs and the minimum stationary points using the second-order González–Schlegel integration method [103,104]. Solvent effects of THF were considered by the full optimization of the gas-phase structures at the same computational level using the polarizable continuum model (PCM) [70] in the framework of the self-consistent reaction field (SCRF) [71,72].

The GEDT [75,76] values were estimated by natural population analysis (NPA) [53,54] using the equation GEDT(f) = charge q_f_, where q is the atoms of a framework (f) at the TSs. Indices of single-bond development (l) were estimated according to the equations recommended in reference [73,74].

NPA and ELF studies were performed with TopMod 09 [105] software. For the visualization of the molecular geometries of all structures and 3D representations of radical anions and radical cations, GaussView 6.0 software [106] was used. In turn, the ELF localization domains were represented by using ParaView 5.9.1 software [107] at an isovalue of 0.75 a.u.

The molecular docking for the obtained (4*RS*,5*RS*)-3-aryl-4-nitro-5-tribromomethyl-2-isoxazolines (**3a-e**) was performed by Gnina 1.3 software [108]—a fork of smina (itself being a fork of Autodock Vina [109]) that uses scoring with convolutional neural networks. The docking mode was chosen to be flexible and performed in two batches—all residues within the radiuses of 5.0 Å and 3.5 Å from the native-ligand position were considered to be flexible. The docking was performed 5 times for (4*R*,5*R*)- and (4*S*,5*S*)-2-isoxazolines (**3a-e**) for each of the 3 proteins using random seeds each time, resulting in 125 docking scores in total for each batch.

The ligands studied were chosen to be autoboxed to the coordinates of the native ligand, i.e., the existing ligand from the original RSCB PDB database [110] protein file. The ensemble of neural network scoring models was chosen to be the default, the maximum number of conformers kept was set to default 9, and the level of exhaustiveness was set to 64.

## 4. Conclusions

Our comprehensive experimental and computational study sheds light on the formation of tribromomethyl-substituted analogs of 4-nitro-2-isoxazoline. We found that all considered processes of these compounds group were realized under mild conditions, with full regioselectivity and with maintaining the primary geometrical isomerism of the tribromomethyl group and nitro group. In all evaluated reactions, only (4*RS*,5*RS*)-3-aryl-4-nitro-5-tribromomethyl-2-isoxazolines were formed. The individuality and the constitution of the obtained compounds were confirmed without any doubt on the basis of the chromatographic experiments and spectral analysis. As a consequence, we can propose a method for the effective and selective preparation of unique analogs of 4-nitro-2-isoxazoline functionalized by the sterically tribromomethyl group. It should also be highlighted that the observed regioselectivity is explained by the nature of the local electrophile–nucleophile interactions determined by the electronic interactions of used reagents.

The analysis of the computed Eyring parameters confirms the formation of (4*RS*,5*RS*)-3-aryl-4-nitro-5-tribromomethyl-2-isoxazolines as the most preferable product from a kinetic point of view. On the other hand, the exploration of the reaction profiles shows that the mechanism of the formation of the isoxazoline ring is not stepwise, as we had expected on the basis of the examples from the recent literature. This is not a synchronical Huisgen-type, single-step mechanism, but the *two-stage*, *one-step* asynchronous mechanism, in which the formation of the O-C(CBr_3_) bond takes place once the C-C(NO_2_) bond is already formed.

At the end, the obtained (4*RS*,5*RS*)-3-aryl-4-nitro-5-tribromomethyl-2-isoxazolines (as separate (4*R*,5*R*) and (4*S*,5*S*) isomers) were docked to three proteins: gelatinase B, COX-1, and Caspase-7. Based on the obtained results, it can be concluded that gelatinase B seems to be the best target for (4*R*,5*R*)-3-(4-chlorophenyl)-4-nitro-5-tribromomethyl-2-isoxazoline, which has the most promising docking scores, whereas (4*R*,5*R*)-3-(4-methoxyphenyl)-4-nitro-5-tribromomethyl-2-isoxazoline scores the best for both for COX-1 and Caspase-7. In these cases, the docking scores are significantly lower, but the obtained results suggest that the mentioned molecule is a more valid ligand for cyclooxygenase COX-1 than for Caspase-7.

For (4*S*,5*S*)-2-isoxazoline, the energy score is different. Although the best binder to gelatinase B turned out to be (4*S*,5*S*)-3-(4-chlorophenyl)-4-nitro-5-tribromomethyl-2-isoxazoline as well, for COX-1 and Caspase-7, the best scorers were (4*S*,5*S*)-3-(4-methylphenyl)-4-nitro-5-tribromomethyl-2-isoxazoline and (4*S*,5*S*)-3-(4-chlorophenyl)-4-nitro-5-tribromomethyl-2-isoxazoline yet again. We hope that the presented study will be helpful for determining future directions of application for this class of organic compounds.

## Data Availability

The data presented in this study are available on request from the corresponding authors.

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
