# Peer review of "A Comprehensive Study of the Synthesis, Spectral Characteristics, Quantum–Chemical Molecular Electron Density Theory, and In Silico Future Perspective of Novel CBr3-Functionalyzed Nitro-2-Isoxazolines Obtained via (3 + 2) Cycloaddition of (E)-3,3,3-Tribromo-1-Nitroprop-1-ene"

_molecules, 2025, doi:10.3390/molecules30102149_

Round 1
Reviewer 1 Report
Comments and Suggestions for Authors
A Comprehensive Study of Synthesis, Spectral Characteristic, Quantum Chemical MEDT and In-Silico Future Perspective of Novel CBr3-functionalysed Nitro-2-isoxazolines Obtained via (3 + 2) Cycloaddition of (E)-3,3,3-Tribromo-1-nitroprop-1-ene
Jasiński and co-workers report the systematic experimental and quantum chemical studies of the cycloaddition product of substituted ANOs and TBNP. The MEDT and CDFT studies are well-described to understand the charge distribution and nucleophilic and electrophilic centers in the cycloaddition components. The authors then present the experimental section with the synthesis and characterization studies. The theoretical study based on Potential Energy Surface (PES) as well as Bonding Evolution Theory (BET) supports the formation of the final product 3a-3e. The authors have also explored the biological activity of cycloaddition products as enzyme inhibitors and ligands via activity spectra (PASS) and molecular docking studies.
The article is of considerable interest to the readership of the Molecules journal. Therefore, the reviewer recommends accepting pending minor revisions to address the outlined concerns.
Here are some points to consider for revision:
- On Page 4, in the ELF analysis, when the p-position of the ANO phenyl ring is substituted with different electronegative and electropositive groups, what are the respective electron population distributions?
- What does “÷” in line 155 mean?
- In section 2.2.1, the characterization (NMR, FT-IR, etc.) of the cycloaddition components (1a-e) and (2) should be provided.
- HSQC spectrum should be provided for the assignment of the stereochemistry of the double bond in TBNP.
- In section 2.2.1, abbreviations such as NCS, DMFA, etc. should be mentioned next to their expanded names in the text for a better understanding of Schemes 2 and 3.
- In Section 2.4.1, what is the deciding factor for these enzymes– ubiquinol-cytochrome-c reductase and glucan endo-1,6-beta-glucosidase? What other classes of enzymes can be tested for inhibition?
- In section 3.1, bromal should be mentioned in the parentheses of tribromoacetaldehyde.
- In section 3.2, in FT-IRs, mention the Ar-X stretching frequencies in compounds 3a-3e.
Reviewer 2 Report
Comments and Suggestions for Authors
Comments for the Authors
In this review article, the authors have explored an intriguing subject that has been somewhat overlooked in recent years, A Comprehensive Study of Synthesis, Spectral Characteristic, Quantum Chemical MEDT and In-Silico Future Perspective of Novel CBr3-functionalysed Nitro-2-isoxazolines Obtained via (3 + 2) Cycloaddition of (E)-3,3,3-Tribromo-1-nitroprop-1-ene. Their focus on the recent advancements in this area offers valuable insights into Dementia and more. Considering the limited availability of such articles, this manuscript holds considerable potential for publication in Molecules. However, further polishing is required to meet the journal's standards. With its concise content, this article promises to offer valuable insights to scientists. After undergoing no revisions, I endorse the publication of this manuscript in Molecules, where it can contribute significantly to the field.
- The manuscript was well written and but can improve the English further.
- Line 120, authors should comment on it. What does the ELF topological analysis of BNO (1) support its classification as a propargyl-type zwitterionic TAC ?
- Followup to the above one, how is the significance of the electron population in its monosynaptic and disynaptic basins?
- Line 122, authors should comment on it. Is there any differences in electron population distribution were observed between BNO (1) and TBNP (2) according to the ELF analysis, and
- Followup to the above one, If yes, what does these findings correlate with their respective bond characterizations?
- Line 164, Need more explanation, following the computed global electrophilicity (ω) and nucleophilicity (N) indices, what does the electronic properties of ANOs (1a–e) compare to those of TBNP (2), and how does these values suggest about their reactivity in polar reactions?
- Line 165, what is ANO (1e) classified as a strong electrophile compared to ANOs (1a–d), and
- Followup to the above one, what does the presence of different substituents influence the ω and N indices?
Comments for the Authors
In this review article, the authors have explored an intriguing subject that has been somewhat overlooked in recent years, A Comprehensive Study of Synthesis, Spectral Characteristic, Quantum Chemical MEDT and In-Silico Future Perspective of Novel CBr3-functionalysed Nitro-2-isoxazolines Obtained via (3 + 2) Cycloaddition of (E)-3,3,3-Tribromo-1-nitroprop-1-ene. Their focus on the recent advancements in this area offers valuable insights into Dementia and more. Considering the limited availability of such articles, this manuscript holds considerable potential for publication in Molecules. However, further polishing is required to meet the journal's standards. With its concise content, this article promises to offer valuable insights to scientists. After undergoing no revisions, I endorse the publication of this manuscript in Molecules, where it can contribute significantly to the field.
- The manuscript was well written and but can improve the English further.
- Line 120, authors should comment on it. What does the ELF topological analysis of BNO (1) support its classification as a propargyl-type zwitterionic TAC ?
- Followup to the above one, how is the significance of the electron population in its monosynaptic and disynaptic basins?
- Line 122, authors should comment on it. Is there any differences in electron population distribution were observed between BNO (1) and TBNP (2) according to the ELF analysis, and
- Followup to the above one, If yes, what does these findings correlate with their respective bond characterizations?
- Line 164, Need more explanation, following the computed global electrophilicity (ω) and nucleophilicity (N) indices, what does the electronic properties of ANOs (1a–e) compare to those of TBNP (2), and how does these values suggest about their reactivity in polar reactions?
- Line 165, what is ANO (1e) classified as a strong electrophile compared to ANOs (1a–d), and
- Followup to the above one, what does the presence of different substituents influence the ω and N indices?
Reviewer 3 Report
Comments and Suggestions for Authors
Check line 174 to see if it is higher or lower.
From the energy of the frontier orbitals (HOMO/LUMO), check if the reaction between ANOs and TBNP is a normal electron-damand and inverse electron-damand reaction.
With the infrared (IR) and electronic circular dichroism (ECD) spectroscopic study, they can simulate (4R,5R) or (4S,5S) configurations of 32CAs products.
TS of Figure 5. 3a is the stereoisomer (4R,5R) justify why TS stereoisomer (4S,5S) not carried out, the same for 4a
In the Molecular Docking with which stereoisomer (4R,5R) or (4S,5S) was it carried out, and which one has better affinity values
Reviewer 4 Report
Comments and Suggestions for Authors
MS N° ID: molecules-3588968 by Karolina ZawadziÅ„ska-Wrochniak et al.
The present manuscript reports the synthesis of five different (4SR,5RS)-3-aryl-4-nitro-5-tribromomethyl-2-isoxazolines. The authors describe aspects of the molecular mechanism of the synthetic procedure and verify a regioselectivity of the process.
The synthetic processes, which, however, follow procedures already described in the literature for such types of compounds, are adequately described. The chemical-physical characterization of the molecules is also adequate.
The authors report in the manuscript, in addition to the synthetic experimental and chemical-physical characterization procedures, some theoretical evaluations that are derived from in silico computational approaches or obtained through the PASS program.
The authors conduct with the PASS program an initial theoretical evaluation of potential and generic antimicrobial activity (antiviral, antibacterial, antifungal). Here already erroneously reported as antimicrobial is the antiparasitic activity, which is instead to be ascribed to non-microscopic-sized pathogens.
This generic antimicrobial activity refers to the lack of identification of specific microorganisms belonging to the categories of bacteria, fungi or viruses are concerned. In addition, no information is obtained on potential targets for the molecular mechanism of efficacy of the molecules.
Upon careful evaluation of the text what emerges is the lack of proof of concept related to the putative biological activities of the compounds.
In fact, there are no data - even preliminary in vitro data (such as binding or in vitro evaluations of the actual inhibition constants of the enzyme systems that the authors refer to). The evaluations are exclusively predictive and theoretical.
The authors, although the title does not make references to specific biological activity, in the description of the experimental and theoretical research conducted - with specific references to the abstract and conclusions sections, for example - affirm that molecules may have a specific biological potential. This claim, of specific and substantial relevance, must be supported by experimental data.
Then the rationale with which the theoretical studies were conducted is not understood.
It is clear that any compound can exhibit a multitarget interaction, with greater or lesser affinity for different targets. Instead, the authors in the present manuscript shift the focus of discussion and potential biological activity by moving from one target to another without a rational and detailed description.
Moreover, as example, the results obtained from predictive analysis through the PASS analysis, are further tentatively confirmed by in silico studies. It would be appropriate to justify the choice of not carrying out molecular docking studies on the targets identified as potentially relevant by the predictive analysis. In the list of targets envisaged by PASS analysis,, it would be useful to specify the Pa (probability of activity) values related to the interaction with COX-1, caspase-7 and gelatinase B, if present.
Since a possible interaction with COX-1 has been shown, a potential anti-inflammatory activity, supported by these predictive data, could also be discussed.
Moreover, some minor point emerged:
- Regarding the experimental part, it is suggested that the solvent N,N-dimethylformamide, be indicated with the correct acronym DMF, according to the IUPAC convention and the most widespread international usage. Finally, it would be helpful to make the synthesis description more detailed, including parameters such as reaction temperatures, reaction times and possibly the yields obtained.
For the reasons cited above I recommend rejection by Molecules for the manuscript in the present form.
